

# Geotechnical controls on erodibility in fluvial impact erosion

Jens M. Turowski[1], Gunnar Pruß[1], Anne Voigtländer[1], Andreas Ludwig[2], Angela Landgraf[2], Florian Kober[2], Audrey Bonnelye[1,3]

[1]Helmholtzzentrum Potsdam, GFZ German Research Centre for Geosciences, Telegrafenberg, 14473 Potsdam, Germany
[2]Nationale Genossenschaft für die Lagerung Radioaktiver Abfälle (NAGRA), 5430 Wettingen, Switzerland
[3]University of Lorraine, Nancy, France

*Correspondence to*: Jens Turowski (jens.turowski@gfz-potsdam.de)

**Abstract.** Bedrock incision by rivers is commonly driven by the impacts of moving bedload particles. The speed of incision is modulated by rock properties, which is quantified within a parameter known as erodibility that scales the erosion rate to the erosive action of the flow. Although basic models for the geotechnical controls on rock erodibility have been suggested, large scatter and trends in the remaining relationships indicate that they are incompletely understood. Here, we conducted dedicated laboratory experiments measuring erodibility using erosion mills. In parallel, we measured compressive strength, tensile strength, Young's modulus, bulk density and the Poisson ratio for the tested lithologies. We find that under the same flow conditions, erosion rates of samples from the same lithology can vary by a factor of up to sixty. This indicates that rock properties that may vary over short distances within the same rock can exert a strong control on its erosional properties. The geotechnical properties of the tested lithologies are strongly cross-correlated, preventing a purely empirical determination of their controls on erodibility. The currently prevailing model predicts that erosion rates should scale linearly with Young's modulus and inversely with the square of the tensile strength. We extend this model using first-principle physical arguments, taking into account the geotechnical properties of the impactor. The extended model provides a better description of the data than the existing model. Yet, the fit is far from satisfactory. We suggest that the ratio of mineral grain size to the impactor diameter present a strong control on erodibility that has not been quantified so far. We also discuss how our laboratory results upscale to real landscapes and long timescales. For both a revised stream power incision model and a sediment-flux-dependent incision model, we suggest that long-term erosion rates scale linear with erodibility and that, within this theoretical framework, relative laboratory measurements of erodibility can be applied at the landscape scale.

## 1 Introduction

Rivers can cut rock, which usually is a slow process (Koppes and Montgomery, 2009; Molnar, 2001), sometimes carving deep canyons over thousands of years (Karlstrom et al., 2014). Yet, fluvial bedrock erosion can also be rapid, with centimeters or even meters of incision within a single flood (e.g., Cook et al., 2013; Hartshorn et al., 2002; Nativ and Turowski, 2020; Turowski et al., 2008; Lamb and Fonstad, 2010). Erosion rates within bedrock rivers result from a competition between driving and resisting forces, quantified by erosivity and erodibility, respectively. Consequently, they can be expected to depend on the





properties of the eroded rock. Rock property control on erodibility has been qualitatively suggested from many morphological field observations. For example, rivers are commonly narrower and steeper in hard rock than they are in soft rock (e.g., Brocard and van der Beek, 2006; Bursztyn et al., 2015; Wohl and Ikeda, 1998; Wohl and Merritt, 2001; Montgomery, 2004). However, empirical joined datasets of erodibility and rock properties that allow a quantitative analyses of rock property controls on

erodibility are rare. In the field, they are notoriously difficult to acquire due to multiple controls on erosion rates that are hard to disentangle, spatial and lithostratigraphic variability, local alterations of the rock due to weathering, weak preservation, or poor exposure. Experimental approaches have been used (e.g., Sklar and Dietrich, 2001, Sunamura et al., 1985), but present challenges in the scaling of flow properties (Attal et al., 2006; Lewin and Brewer, 2002), comparability, and in covering a broad range of different rock types.


The controls of physical rock properties on erodibility can be expected to be specific to a particular erosion process. For impact erosion, it has been suggested that erodibility scales linearly with the substrate's Young's modulus, which describes its elastic response, and inversely with the square of the tensile strength, the maximum tensile force the rock can endure without breaking (Sklar and Dietrich, 2004). This scaling is currently used as the state of the art in theory and experiments (Auel et al., 2017;

Beer and Lamb, 2021; Inoue et al., 2017; Miller and Jerolmack, 2021; Sklar and Dietrich, 2004). However, there are a number of observations and considerations that suggest that it does not give a full description of the observations. For example, the experimental data from the erosion mills of Sklar and Dietrich (2001) and the flume of Inoue et al. (2017) show around an order of magnitude of scatter in erosion rates for rocks with similar tensile strength. In addition, a relationship with tensile strength remains if the data are corrected for the proposed inverse-square relationship (Müller-Hagmann et al., 2020). Young's

modulus is often not measured on the sampled rocks, but estimated from measurements of similar rock types, and it is frequently assumed to vary only little (e.g., Sklar and Dietrich, 2004). Some flume experiments suggest a control of compressive rather than tensile strength on erodibility (Sunamura and Matsukura, 2006; Sunamura et al., 1985), and a negative correlation with Young's modulus has been reported for concrete (Scott and Safiuddin, 2015). In addition, experiments on wind-driven impact erosion have yielded more complicated relationships than are currently used for fluvial processes (e.g.,

Momber, 2016; Verhoef, 1987), despite the similarity in the process physics. Finally, observations from experiments suggest a dependence on mineral grain size (Hobley, 2005). It can also be expected that the geotechnical properties of the impactor play a role, because they affect the fraction of the kinetic energy of the impact that is transferred to the rock (e.g., Dietrich, 1977; Finnegan et al., 2017; Johnson et al., 2009). Neither of the latter effects is accounted for in current models at the moment. As a result, erodibility and its geotechnical controls remain poorly quantified.


Here, we describe dedicated experiments to shed light on rock property controls on erodibility in fluvial impact erosion. We measured erosion rates for a range of lithologies in mills specifically designed to hold the erosivity of the flow constant (Turowski et al., 2023). In parallel, we recorded geotechnical properties that have previously been suggested to control





erodibility. We evaluate the observations in the light of existing theory based on brittle fracture, and further develop this theory
using first-principle physical arguments.

## 2 Methods and Materials

Here, we give an overview over the methods and sample materials. A detailed description of the mill experimental protocol
has been given by Turowski et al. (2023). All data are available through the publication by Pruß et al. (2023).

### 2.1  Sample sourcing and preparation

Rock samples were collected in northern Switzerland, complemented by few samples from the more southern, alpine region,
and southern Germany, covering a broad range of rocks from 18 lithologic units (Fig. 1, Table 1). Most of the rocks are
sedimentary, including mudstones, sandstones and limestones, but some crystalline rocks are also included. Within
heterogeneous sedimentary units, typically the harder beds were sampled, since weaker sequences (e.g. marls, mudstones)
were difficult to impossible to be properly drilled. Cores were drilled with a water-cooled, 200 mm diamond core bit, if
necessary broken free from the bedrock with a chisel, and lifted out of the hole. Dowels were placed into the top face of the
specimen to hoist it out of the borehole when required. The obtained core diameter varied between the lithologies and ranged
from 191 mm to 193 mm. In some cases, cores were sourced from blocks that were already detached from the bedrock, either
in situ or after transport to the laboratory. Some sites allowed sampling of two different units (e.g., GR, J, see Table 1). We
therefore distinguish between a letter-based site id (e.g., GR), a numerical unit id, which follows the alphabetical order of the
site ids, and a sample id, which is the combination of the site id and a number identifying the core (e.g., GR1, J3, see Table 1).
At some locations, multiple cores of the same unit were collected, either to cover variations of the rock in grain size or
composition, or to obtain cores both parallel and perpendicular to major structural planes, such as bedding planes. The cores
were subsampled for erodibility and rock property measurements as described below.



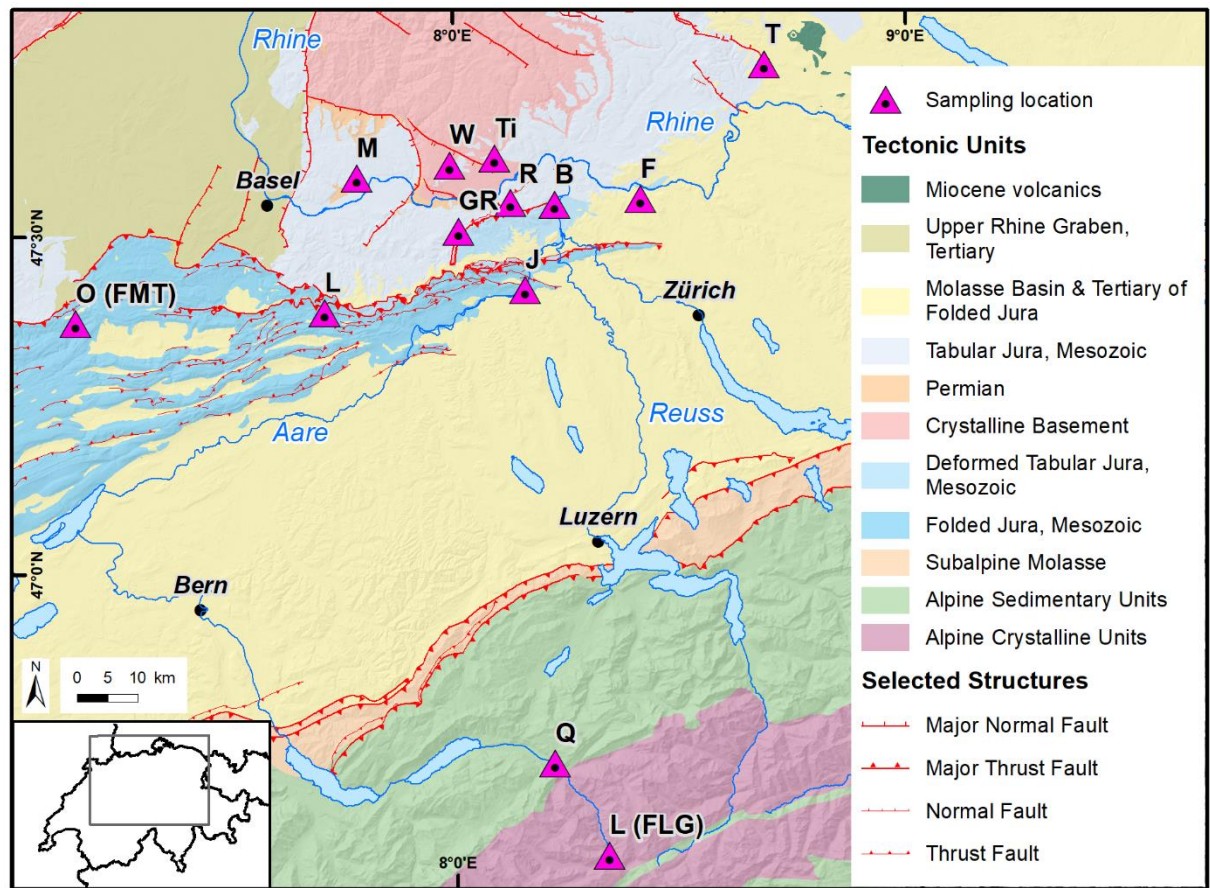


**Figure 1: Overview map with sampling locations. Sampling locations are labelled as detailed in Table 1. The tectonic map is based on Nagra (2014), showing a subset of the structures in the current display (selected main and regional thrust and normal faults at surface). Differentiation of sedimentary and crystalline units in the alpine region has been added following Kühni and Pfiffner (2001). The background digital terrain model corresponds to the Shuttle Radar Topography Mission (SRTM) 1 Arc-Second Global (NASA 2016).**


With the exception of the Opalinus Clay samples (Unit 12), which was too soft, all cores were cut with a water-cooled diamond saw to obtain discs with a thickness of about 55 mm for the mill experiments (mill samples) and 120 mm for geotechnical testing (core samples). If possible, multiple discs were produced from the same core. Subsequently, both faces of the mill samples were ground to a length of 50 mm, to ensure that they were planar, parallel, and had a comparable surface roughness.

At least one core sample was cut for each of the lithological units except the Passwang Formation due to insufficient core dimensions, and the Opalinus Clay, because its clay-rich structure means that standard protocols for geotechnical testing do not work (Giger et al., 2018; Minardi et al., 2021).

To obtain samples for compressive and tensile strength testing, the 120 mm core samples were subsampled to obtain cores with 50 mm diameter, yielding a maximum of eight samples from each core segment. Half of these were designated for 100 compressive strength testing and half for (indirect) tensile strength testing. For tensile strength testing, the samples were further





cut to obtain discs of 25 mm thickness. For compressive strength testing, the cores were first cut to a length of 104 mm and then ground on both sides to obtain a length of 100 mm, and to ensure planar and parallel faces.

**Table 1: Sites, units and samples used for the experiments. For further details on lithostratigraphic units see Jordan & Deplazes (2019), unless otherwise indicated. Lithostratigraphic names are used according to the specified references.**

| # | Lithological unit | Rock type | Location | Location id | Core id |
|---|---|---|---|---|---|
| 1 | Passwang Formation | (Sandy) limestone | Böttstein | B | B1, B2 |
| 2 | Lower Freshwater Molasse Group | Sandstone | Fisibach | F | F1, F4, F5 |
| 3 | Klettgau Formation, Gansingen Member | Dolomite | Gruhalde | GR | GR1 |
| 4 | Staffelegg Formation, Beggingen Member | Limestone | Gruhalde | GR | GR2 |
| 5 | Hauptrogenstein | (Oolitic) limestone | Jakobsberg | J | J1, J7 |
| 6 | Wildegg Formation | Limestone | Jakobsberg | J | J2, J3 |
| 7 | Schinznach Formation, Liedertswil Member | Limestone | Liedertswil | L | L1 |
| 8 | Schinznach Formation, Stamberg Member | Dolomite | Liedertswil | L | L2 |
| 9 | Central Aar Granite (Nagra, 2019) | Granite | Felslabor Grimsel (Alps) | L (GTS) | L206 |
| 10 | Grimsel Granodiorite (Nagra, 2019) | Granodiorite | Felslabor Grimsel (Alps) | L (GTS) | L502 |
| 11 | Oberer Muschelkalk (LGRB, 2021) | Limestone | Minseln (Baden-Württemberg, Germany) | M | M1, M2 |
| 12 | Opalinus Clay | Mudstone | Felslabor Mont Terri | O (FMT) | O1, O2 |
| 13 | Quinten Formation (Gisler et al., 2020) | Limestone | Lammi (Alps) | Q | Q1 |
| 14 | Klettgau Formation, Ergolz Member | Sandstone | Röt | R | R1 |
| 15 | "Massenkalk" | Limestone | Thayngen | T | T1, T6 |
| 16 | "Felsenkalke" | Limestone | Thayngen | T | T2, T4 |
| 17 | Albtal-Granit (LGRB, 2021) | Granite | Tiefenstein (Baden-Württemberg, Germany) | Ti | Ti |
| 18 | Murgtal-Gneisanatexit-Formation (LGRB, 2021) | Gneiss | Wickartsmühle (Baden-Württemberg, Germany) | W | W |



## 2.2 Mill erosion experiments

### 2.2.1 Mill design

Our erosion mills, simulating the motion of bedload particles over a bedrock bed in a river, were specifically designed for the
project (Turowski et al., 2023), based on devices previously described in the literature (Sklar and Dietrich, 2001; Scheingross et al., 2015; Small et al. 2015). While erosion mills do not faithfully produce the flow patterns in mountain streams during floods (Attal et al., 2006), they provide the advantage of easy handling and low costs, a homogenous experimental environment, and a tight, direct control on experimental conditions via only a small number of control variables. We utilized these advantages to construct experimental devices that fulfil four main priorities in the design (Turowski et al., 2023): (i) keeping erosivity
within the mills as constant as possible, (ii) simple and cheap construction to allow easy reproduction, (iii) easy handling and a straight-forward experimental protocol, and (iv) avoiding the need of special equipment, infrastructure or fixtures. The mills are made from acrylic polymer (polymethyl methacrylate, PMMA), which is impervious to corrosion, sufficiently tough, and allows visually monitoring flow patterns and turbidity changes caused by the suspension of the erosion products. The dimensions of the mill are 208 mm in internal diameter and 228 mm in height. While in operation, the three main parts of the
mill - wall, base plate and lid - are clamped together with four threaded rods and knurled screws (Fig. 2). An electrical engine is placed 50 mm above the center of the lid and connected to the stainless steel propeller shaft with a rigid clutch. The opening for the propeller shaft is protected with a seal ring and the three-bladed brass propeller is placed at a height of 153 mm above the mill bottom, i.e. about 100 mm above the initial surface of the sample. The propeller has an outer diameter of 70 mm and a pitch of 71.7 mm. A detailed description of the mill design including technical drawings and experimental protocols has been
given by Turowski et al. (2023).





**Figure 2: Erosion mills in use for experiments, with specimens from different lithologies. The turbidity indicates the concentration of erosion products in the water. Top row: Massenkalk (Unit 15) T1-1B, Lower Freshwater Molasse (Unit 2) F4-1A, Staffelegg Formation, Beggingen Member (Unit 4) GR2-1A, Murgtal-Gneisanatexit-Formation (Unit 18) W-4A. Bottom row: Schinznach Formation, Stamberg Member (Unit 8) L2-1A, Passwang Formation (Unit 1) B1-1A and B2-1A, Klettgau Formation, Ergolz Member (Unit 14) R1-3A.**

### 2.2.2 Experimental protocol

Before the experiment, the samples had to be saturated with water. Otherwise, the gain of mass by uptake of water would conceal or even exceed the loss of mass due to erosion. To saturate the material, the samples were placed in LDPE zipper storage bags with about 1.8 litres of tap water. Trapped air was removed and the samples were stored in light-proof boxes to inhibit the growth of microorganisms like algae. In total the soaking procedure lasted for at least 14 days. Samples were



regularly weighed to a precision of 0.1 g and considered to be saturated when their mass was the same in two successive
weighings.

As abrasive tools in the mills, we used glass beads originally designed for the grinding of pigments. For each experiment two independent bead sets of 150 g each were prepared to run in alternation. To keep track of bead abrasion, after each run the bead set was oven-dried for 24 hours at 40°C, after which each bead was weighed to a precision of 0.01 g to obtain the total weight of the bead set. Wear was compensated for by exchanging glass beads or adding new ones.

Each experiment, for a given sample, consisted of six runs with identical experimental conditions to constrain the measurement error and to track the constancy of erosivity during experiments. During the experiments, the propeller speed was set to 1000±10 rotations per minute. Run duration was set depending on the erosion rates to between 4 hrs and 52 days to achieve a total mass loss of 1 - 10 g. The turbidity of the mill water and prior general experience were used as indicators to set a suitable run duration for the first run of a given sample. After each run, the mill was opened, the sample rinsed, and the water was
exchanged. To measure sample erosion, the water was filtered using 0.2 μm filter paper, the captured material was dried for at least 24 hours at 40°C, and the dry solids were weighed to a precision of 0.01 g and corrected for glass bead abrasion. The mean erosion rate of all six runs is used as a representative value for the experiment, the standard error of the mean as a measure of the uncertainty.

The erosion measurements of the Opalinus Clay (Unit 12), a clay-rich mudstone, provided challenges that did not exist for the
other rock types. In our standard protocol, the rock specimens were saturated with water prior to experiments. However, the Opalinus Clay quickly swells up when wetted, and loses structural integrity as a result (cf. Thury, 2002). As a consequence, it was not possible to saturate the sample with water before the measurements, and the corresponding part of the protocol was skipped. Instead, the runs were immediately started after placing the unsoaked specimens into the erosion mill and adding water.

**2.3 Geotechnical measurements**

For the present study, we recorded tensile strength, compressive strength, Young's modulus, Poisson ratio, and bulk density using standard protocols (DGGT, 2008; Mutschler, 2004) using a MTS Load Frame 315.03 equipped with a load cell 661.31 (1000 kN).

Bulk density was measured from the samples cut for compressive and tensile strength measurements. We measured the height in four positions around the rim, at approximately 90° to each other, with a digital caliper to a precision of 10 μm. Similarly, we measured the diameter in two positions, perpendicular to each other. We used the average of these measurements as representative for height and diameter, and calculated the volume of the sample using the equation for the volume of a cylinder. Sample mass was measured on a digital scale with a precision of 0.1 g, and the bulk density was obtained as the ratio of mass
to volume, assuming a cylindrical geometry.

Uni-axial compressive strength (UCS) was measured following ISRM standards (Mutschler, 2004), with a constant conversion rate of 0.001 mm/s. Compressive strength was assumed to be the maximal recorded pressure before fracture.

Tensile strength was measured using the Brazilian nut splitting test (BZL). In general, we followed the recommendations for sample preparation and protocol (DGGT, 2008). However, it was not possible to test under constant force. Instead, we tested with constant convergence speed of 0.4 mm/min. Tensile strength was calculated as the ratio of twice the maximum force recorded before fracture divided by the volume of the sample (DGGT, 2008).

The static Young's modulus is equal to the slope of the stress-strain plots of the unconfined compressive strength measurements. We used the tangent method to calculate the slope at half the maximum stress recorded before failure.

The Poisson ratio is the ratio of the axial and circumferential length change recorded with strain gauges during the compressive strength experiments. It was calculated as the negative ratio of the slopes of the axial and lateral stress-strain curves. We

prepared a total of 89 samples with strain gauges. Of these, 65 yielded usable data, between 1 and 4 for each lithological unit apart from the Passwang Formation (Unit 1) and the Opalinus Clay (Unit 12), for which no suitable samples were available. We used strain gauges of the type FCAB-6-11 by Tokyo Measuring Instruments Lab. As for Young's modulus, we used the tangent method to calculate the slope at half the maximum stress recorded before failure.

To compare rock properties to mill erosion rates, if available, we used average rock property values from the same core as the mill sample. If no rock property values were available from the same core, we used the average for the lithological unit. The standard error of the mean was used as a measure of uncertainty. Errors of compound quantities were calculated using Gaussian error propagation.

## 3   Results

**3.1 Erosion rate measurements**

Erosion rates varied over approximately six orders of magnitude across all of the tested samples (Fig. 3). The erosion rates measured on samples from the same unit showed some variability, best seen for the six samples of the Lower Freshwater Molasse (Unit 2) with variabilities of up to nearly two orders of magnitude, but also for the Wildegg Formation (Unit 6) and the Quinten Formation (Unit 13), with a variability of more than one order of magnitude (Fig. 3b).




**Figure 3:** Measured mill erosion rates for all samples (A) and lithological units (B). Gray boxes show the median (central horizontal line), and the 25th and 75th percentiles (bottom and top of the box). White squares show the mean, and whiskers the maximum and minimum erosion rates. The display follows the order of Table 1 (core ids, numerical unit id). Sample ids are composed of the leading core id, followed by a letter indicating the position of the sample within the core. Measured erosion rates vary over approximately six orders of magnitude across the different lithologies.

For most of the lithologies, mill erosion rates were comparable over the six runs (Figs. 3A, 4). The standard error of the mean ranged between 1.2% and 35% of the measured erosion rate, with a mean value of 8.5% and a median value of 4%. Uncertainties scale with measured erosion rates (Figs. 4, 5). The results indicate that erosivity was constant in the experiments and the variation in erosion rates across samples is due to variations in erodibility (Turowski et al., 2023). Erosion rate can therefore be used as a proxy for relative erodibility. Erosion rates can vary substantially for samples from the same core, and for cores from the same lithological unit. For example, erosion rates measured on six samples from the Lower Freshwater Molasse (Unit 2), with two samples each cut from three different cores, show minimum and maximum erosion rates of $(0.20\pm0.01)$ g/h and $(12.40\pm1.03)$ g/h, respectively (Fig. 4), which corresponds to a factor of about 63. For a single core, we




see a maximum deviation of a factor of 2.5 in core F1. Other units for which values of several cores were measured yield maximal deviation factors of 2.3 for the Hauptrogenstein oolitic limestone (Unit 5), 11.6 for the Wildegg Formation limestone (Unit 6), 1.3 for the Stamberg Member of the Schinznach Formation, a dolomite (Unit 8), and 1.1 for the Massenkalk limestone (Unit 15). For samples from the same core, we obtained maximal deviation factors of 1.8 for the Hauptrogenstein oolitic limestone (Unit 5), 9.9 for the Quinten Formation limestone (Unit 13), and 1.0 for the Klettgau Formation sandstone (Unit 14).

We conducted experiments with three samples from the Opalinus Clay (Unit 12), two from the shaly facies (O2), and one from the sandy facies (O1). In a pilot experiment on a shaly facies sample, the rock had lost all structural integrity after only 15 minutes of run time. By the end of the experiment, the initially 5 cm thick sample had expanded to a thickness of more than 7 cm. In addition to erosion on the upwards-facing sample face, material had detached from the sides of the sample. Due to the lack of structural integrity, weighing the sample or separating the sediment produced by impact erosion was not possible. The erosion rate could not be measured. For all further experiments with Opalinus Clay, we decided on run times of 2 minutes and slightly adapted the protocol used to empty the mills. Similar problems as in the pilot experiment persisted for the second sample from the shaly facies. Only the first 2-minute run yielded a value of 1037 g/h for the erosion rate. For the sample from the sandy facies, we were able to measure six erosion rate values in runs operated back-to-back. From the six runs, we obtained an erosion rate of 683±107 g/h. In addition to the short run time, swelling, loss of structural integrity due to water uptake and slaking erosion contribute to the uncertainty in the impact erosion measurements of the Opalinus Clay. As a consequence, the uncertainty is large in comparison to measurements on other rock types and cannot currently be quantified. The Opalinus Clay erosion rates thus cannot be directly compared to the other erosion rates and are not considered in the quantitative analysis in the remainder of the paper.

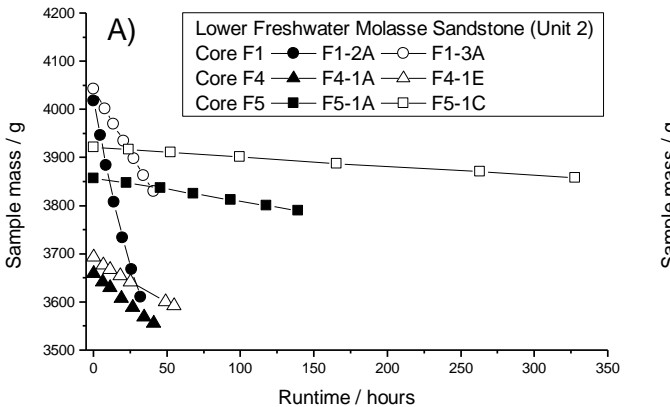
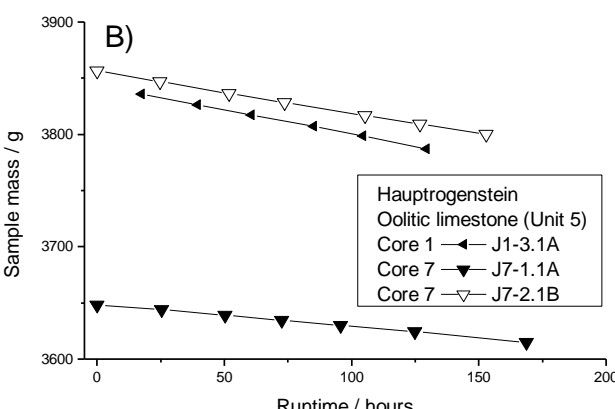

**Figure 4: A) Mass evolution over six runs of six samples from the Lower Freshwater Molasse (Unit 2). Two samples each were tested from three cores, all sourced from the same site. Erosion rates vary considerably, by a factor of up to 63 between cores and up to 2.5 between samples from the same core. B) Mass evolution over six runs of three samples from the Hauptrogenstein oolitic limestone (Unit 5), cut from two different cores. Here, much less variability has been observed, with a factor of up to 2.3 between cores, and 1.8 between samples from the same core.**



## 3.1 Rock geotechnical properties and their relationship to erosion rate

All measured rock geotechnical properties are correlated, in particular compressive strength, tensile strength and Young's modulus, with rank correlation coefficients of 0.83 and above (Fig. 5).

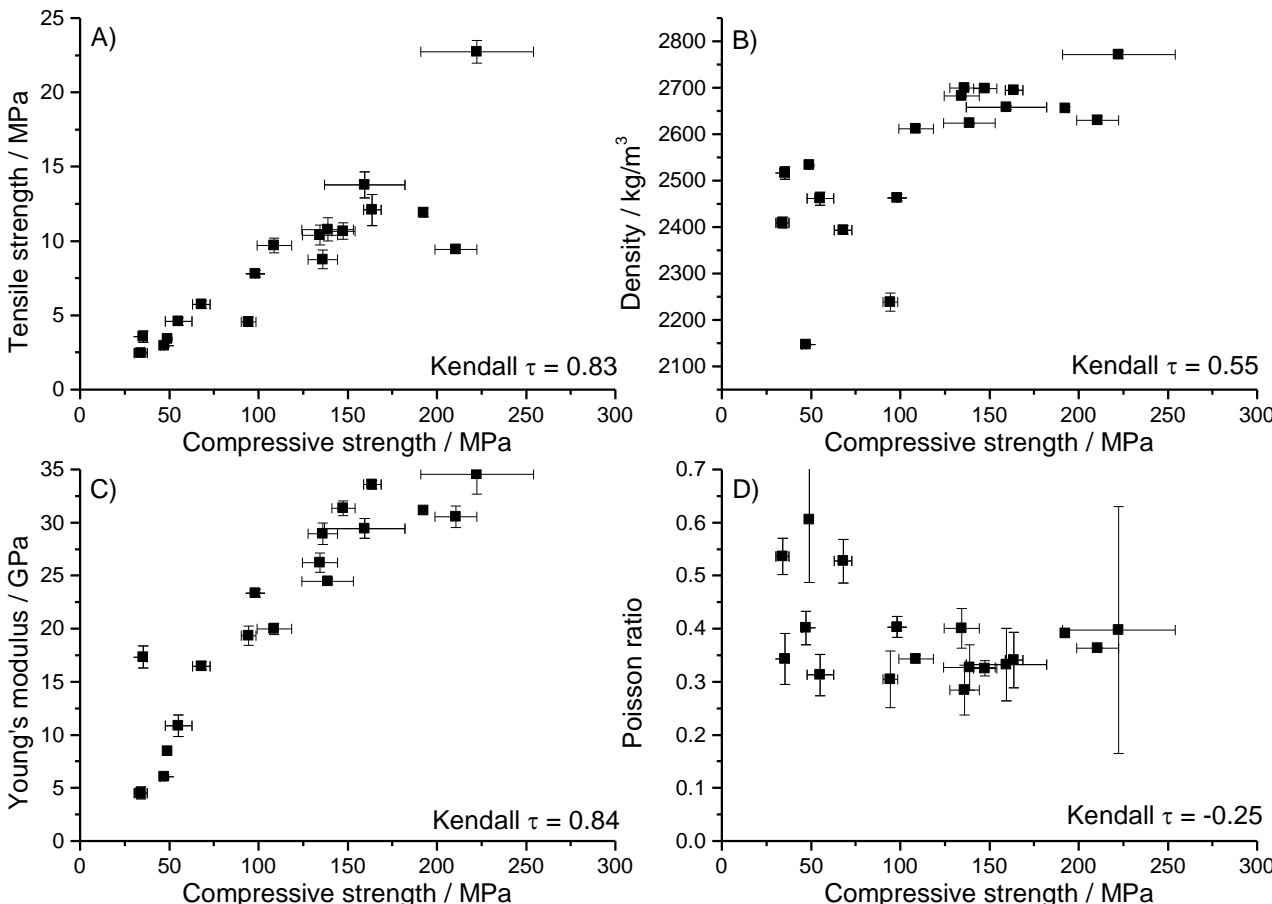

**Figure 5: Relation between rock properties, showing A) tensile strength, B) density, C) Young's modulus, and D) Poisson ratio as a function of compressive strength. All measured rock geotechnical properties are correlated, in particular compressive strength, tensile strength and Young's modulus. Kendall's rank correlation coefficient τ is given on the plots.**

Erosion rate scales with the inverse of rock compressive strength (Fig. 6a). A similar relationship can be observed for rock tensile strength, density, and Young's modulus (Fig. 6b, c, d). However, this seems to be due to the strong correlation between these rock properties (Fig. 5).



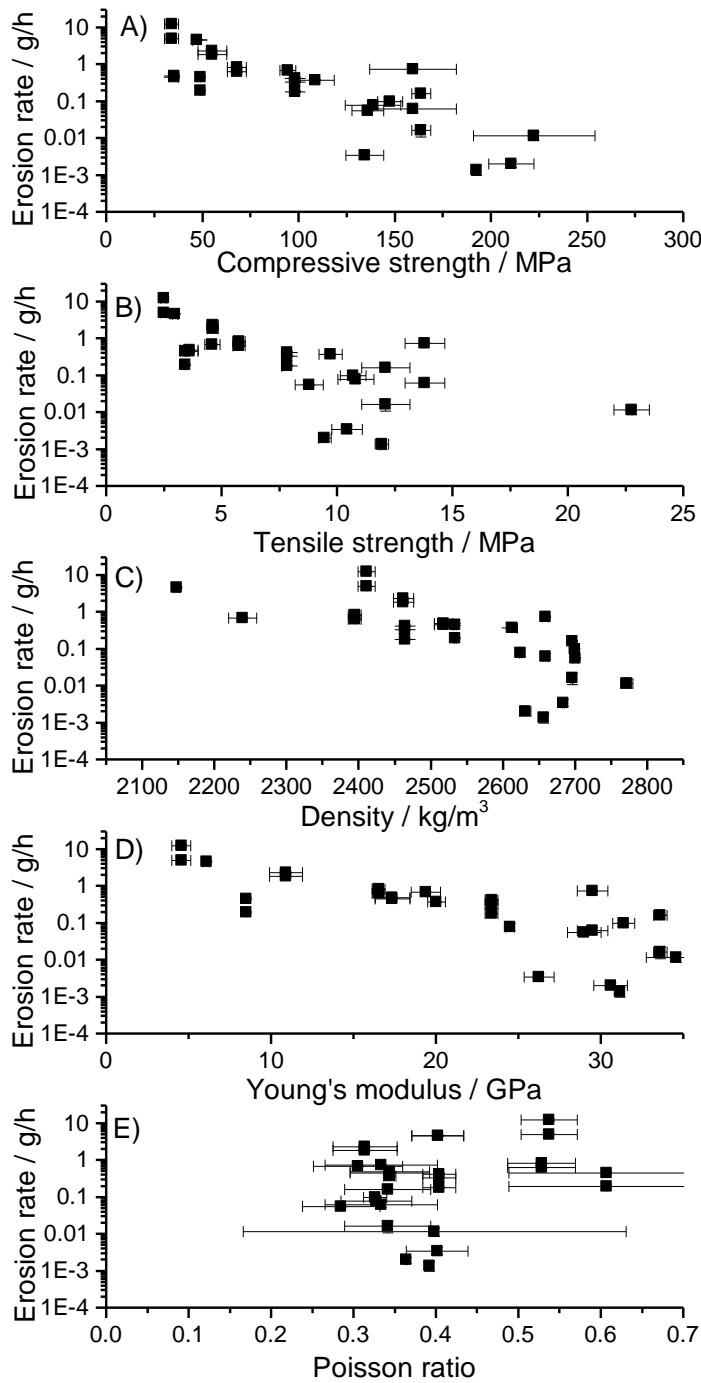

**Fig. 6: Erosion rates as a function of geotechnical properties. Erosion rate scales inversely with compressive strength (A), tensile strength (B), Young's modulus (C) and density (D), and positively with Poisson's ratio (E).**





## 4 Discussion

### 4.1 General remarks and comparison to previous measurements

For different rock types, erosion rates scatter over nearly six orders of magnitude for constant experimental conditions (Fig. 3), with weakly consolidated sandstones or mudstones showing the highest erosion rates, and crystalline rocks such as granite showing the lowest. More remarkable, erosion rates vary by a factor of up to 63 for samples sourced from the same lithological unit, but different cores drawn at the same location, and by a factor of up to ten for samples cut from the same core (Fig. 4).

Measured mill erosion rates plot on a similar trend with tensile strength as previously reported, with lower erosion rates for similar values of tensile strength (Fig. 7). Even though, for a given tensile strength, erosion rates vary by up to three orders of magnitude. The high variability indicates that erodibility is very sensitive at least to some rock parameters other than tensile strength that can vary over short distances within the same lithology, such as grain sizes, mineralogy, cement, local fractures, or flaws.

Generally, our experiments yielded smaller erosion rates for similar tensile strength than has been previously reported. Specifically, the erosion rates measured in our mills are on average only about 6% of the erosion rates measured by Sklar and Dietrich (2001) for rocks with similar tensile strength (Fig. 7). We used a similar mill design as Sklar and Dietrich (2001), similar propeller rotation speed and the same sediment mass of 150 g. The difference thus likely arises from the use of spherical glass beads as abrasive tools, rather than natural quartz pebbles. In addition, some of our rocks show very small erosion rates, 270 pulling down the relationship.

Previously, tensile strength (Beer and Lamb, 2021; Sklar and Dietrich, 2001, 2004; Sunamura and Matsukura, 2006; Müller-Hagmann et al., 2020) and compressive strength (Sunamura et al., 1985) have been suggested to control rock erodibility to impact erosion. Young's modulus has also been suggested as a control (e.g., Sklar and Dietrich, 2004; Scott and Safiuddin, 2015), but is usually assumed to vary within a small range for natural rocks, and has thus not been systematically investigated.

For the rocks tested here, compressive strength, tensile strength, density, Young's modulus, and Poisson's ratio are all correlated to each other (Fig. 5) and to erosion rate (Fig. 6). As a result, it is not possible to empirically distinguish rock property controls on erodibility. Instead, we turn to a theoretical approach to evaluate the relationships.



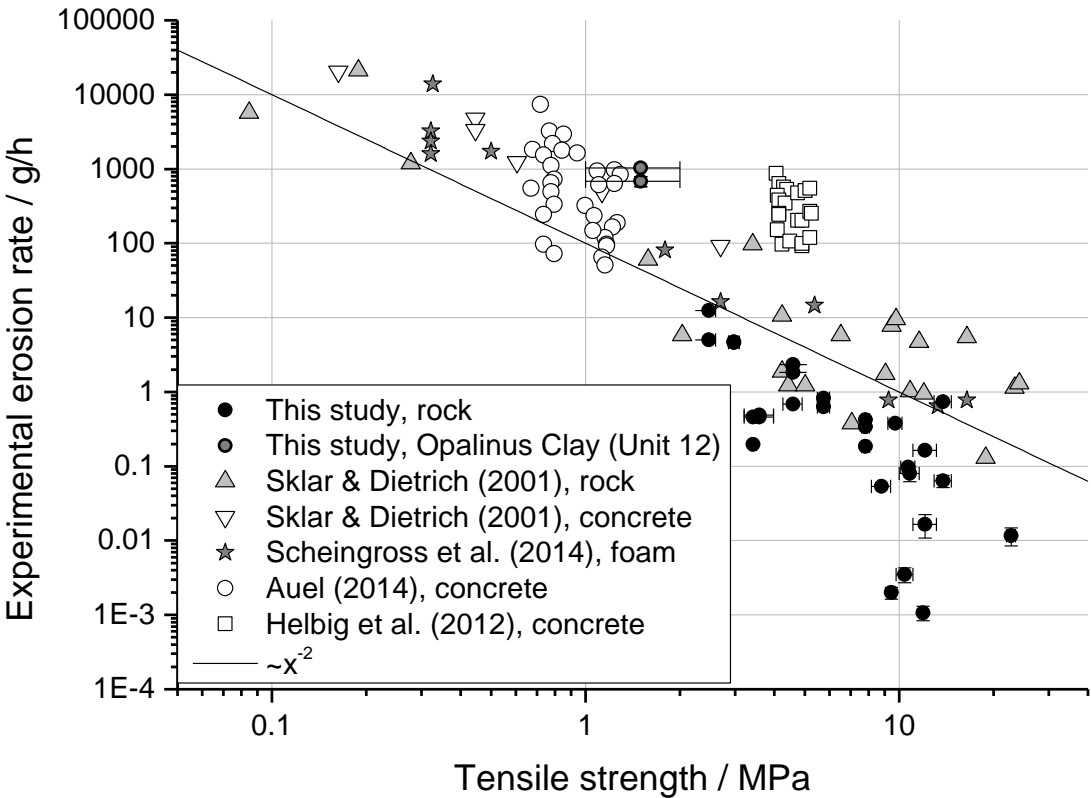

**Fig. 7: Erosion rates measured in the mills (black dots, Fig. 3) in comparison to literature data, showing erosion rates as a function**
**of tensile strength for rock (black circles, grey triangles), concrete (white symbols), and foam (stars). The data were measured in**
**erosion mills by Sklar and Dietrich (2001) on rock (grey triangles) and concrete (white triangles), by Scheingross et al. (2014) on**
**foam (stars), in a linear flume by Auel et al. (2017) on concrete (white circles), and in tumbling mills by Helbig et al. (2012) on**
**concrete (white boxes). For completeness, we added the erosion rates measured for the Opalinus Clay (Unit 12; gray-filled circles),**
**using tensile strength as determined by Bossart and Thury (2008). The solid line indicates the inverse square trend expected from**
**theory.**

## 4.2 Evaluation and extension of the brittle fracture theory

### 4.2.1 Critical elastic energy of the substrate

Following Sklar and Dietrich (2004), who based their arguments on the brittle fracture theory by Engle (1978), we postulate

that fracture upon impact occurs in tension. The eroded volume is assumed to be proportional to the energy delivered to the

rock by impacts, as has been established for impact erosion of brittle materials (e.g., Bitter, 1963; Beer and Lamb, 2021; Miller

and Jerolmack, 2021). Following Sklar and Dietrich (2004), we assume that the erosion rate is the product of the energy

delivered to the substrate per unit area and unit time, which can be expressed as the product of four factors. First, the average

kinetic energy delivered by a single impact $E_{kin}$, second, the impact rate per unit area and time $I_{BL}$, and third, the fraction $f$ of





the kinetic energy that is actually transferred to the rock upon the impact, as tensile elastic energy. These three factors combine
to make up the erosivity $\chi$ of the process. Finally, fourth, the erodibility is the volume eroded per unit energy $\zeta$. Thus, the
erosion rate $E$ is given by:

$$E = \zeta \chi = \zeta f I_{BL} E_{kin}.$$

(1)

The impact rate and kinetic energy have been previously evaluated by Sklar and Dietrich (2004) and Auel et al. (2017), and
will not be further discussed here. The erodibility can be assumed to be inversely proportional to the maximum elastic energy
per unit volume $E_f$ that the rock can experience without fracturing (Engle, 1978). This is proportional to the square of the
fracture strength – here tensile strength $\sigma_T$, because failure occurs in tension – divided by the elastic modulus $Y$:

$$E_f = \frac{1}{2}\frac{\sigma_T{}^2}{Y}.$$

(2)

So, the erodibility is proportional to the inverse of eq. (2):

$$\zeta = \frac{1}{k_{\zeta a}}\frac{Y}{\sigma_T{}^2}.$$

(3)

The rock resistance coefficient $k_{\zeta a}$ has previously been assumed to be constant, but may capture other controls on erodibility
(e.g., Turowski et al., 2013; Auel et al., 2015). If eq. (3) is correct, we expect that trends with geotechnical parameters vanish
when measured erosion rates are normalized by erodibility. This is not the case (Fig. 8), and trends with compressive
strength, tensile strength, Young's modulus, and density remain (see also Müller-Hagmann et al., 2020). This indicates that
there are further rock property controls on erodibility that are not yet accounted for by theory. We attempt to constrain them
using an energy conservation argument in the next section.



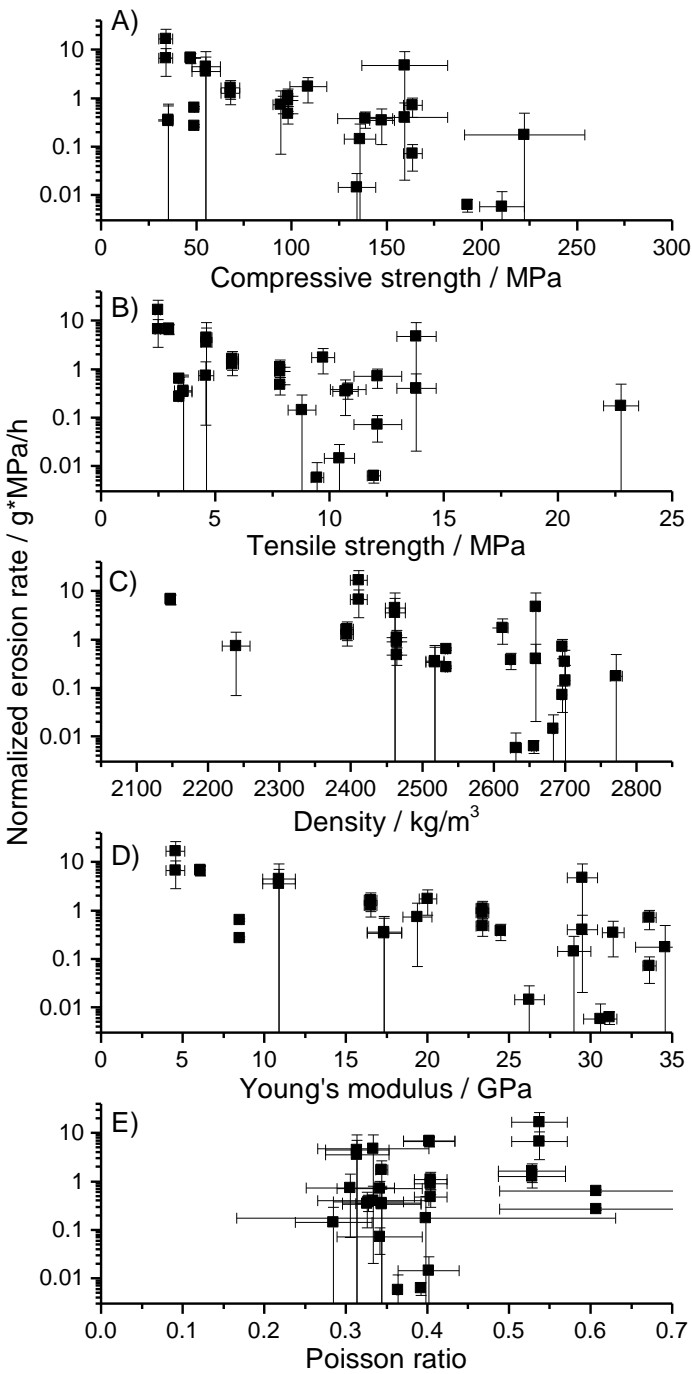

**Figure 8: Erosion rate normalized for the theoretical relationship with tensile strength and Young's modulus (eq. 3) compared to A) compressive strength, B) tensile strength, C) density, D) Young's modulus, and E) Poisson ratio. Trends remain, indicating that current theory is incomplete.**



### 4.2.2 Elastic potential energy in the impactor and substrate

Consider the impact of a bedload particle on the substrate. We are interested in the maximum tensile deformation in the substrate, which, in the context of tensile fracture, has been related to the maximum tensile elastic potential energy in the substrate $E_s$ (eq. 3; see Engle, 1978; Sklar and Dietrich, 2004). This elastic potential energy can be written as the fraction $f$ of the kinetic energy $E_{kin}$ of the impactor at the time of impact, given by the equation

$$E_s = f E_{kin}.$$

325   (4)

Following Sklar and Dietrich (2004), we focus on the vertical component of the impact. Both the particle and the rock deform elastically until all kinetic energy is converted to elastic potential energy. At this point, the particle does not move for an instant, and both the particle and the rock experience the maximum compressive stress due to the impact $\sigma_c$, of equal magnitude and opposite direction. Equating the kinetic and the elastic potential energies of the impactor $E_i$ and the substrate $E_s$, we can

write

$$E_{kin} = E_i + E_s = \frac{1}{2}\frac{\sigma_c{}^2}{Y_i} + \frac{1}{2}\frac{\sigma_c{}^2}{Y} = \frac{1}{2}\left(\frac{1}{Y_i}+\frac{1}{Y}\right)\sigma_c{}^2.$$

(5)

Here, $Y_i$ is Young's modulus of the impactor. Solving eq. (5) for $\sigma_c$ yields

$$\sigma_c{}^2 = \frac{2E_{kin}}{\left(\frac{1}{Y_i}+\frac{1}{Y}\right)}.$$

335   (6)

Using eq. (4), we can also write eq. (5) as

$$E_{kin} = E_i + E_s = \frac{1}{2}\frac{\sigma_c{}^2}{Y_i} + f_c E_{kin}.$$

(7)

Here, $f_c$ is the fraction of the kinetic energy that appears as the maximum compressive elastic potential energy in the substrate.

It is related to $f$ by the square of Poisson's ratio $v$

$$f = v^2 f_c.$$

(8)

By substituting eq. (6) into eq. (7) to eliminate $\sigma_c$, we get

$$E_{kin} = \frac{1}{Y_i}\frac{E_{kin}}{\left(\frac{1}{Y_i}+\frac{1}{Y}\right)} + f_c E_{kin}.$$

345   (9)





After cancelling out $E_{kin}$, we can solve eq. (9) for $f_c$ to give

$$f_c = \frac{Y_i}{Y_i + Y}.$$

(10)

Substituting eqs. (8) and (10) into eq. (1) yields a new expression for the dependence of erosion rate on rock properties


$$E = \frac{1}{k_\zeta} \frac{v^2 Y_i Y}{(Y_i + Y)\sigma_T^2} I_{BL} E_{kin}.$$

(11)

Here, $k_\zeta$ is a rock resistance coefficient. Using eq. (11) on our mill data results in reduced scatter and better fits (Fig. 9), with the adjusted $R^2$ increasing from 0.245 to 0.375.

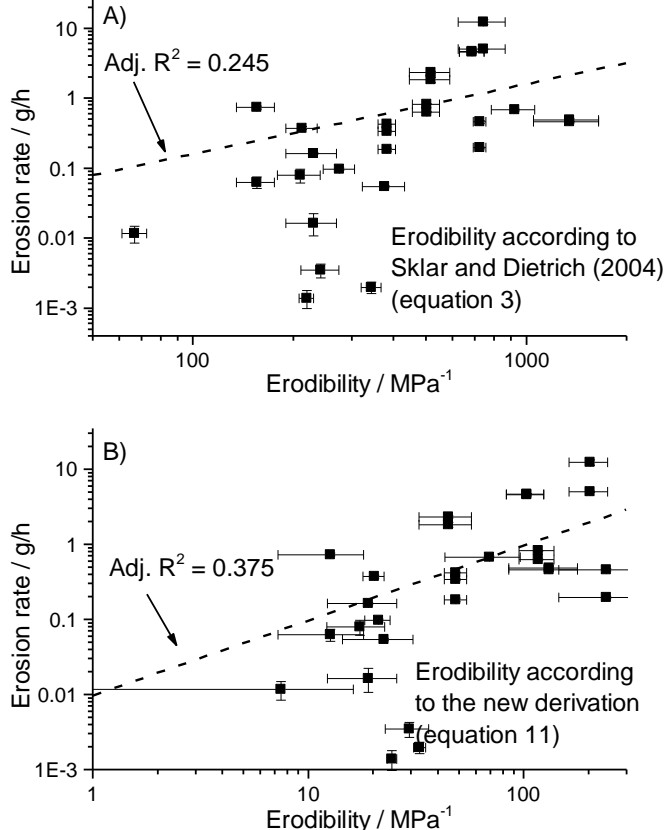

**Figure 9: Erosion rates measured in the mills compared to theoretical predictions. The dashed lines show a linear fit to the logarithmized data, with the slope fixed to one, corresponding to a proportional relationship. A) Erodibility according to Sklar and Dietrich (2004), based on Engle's (1978) theory of brittle fracture (eq. 3). The fit gives an adjusted $R^2$ of 0.245 and a prefactor of $1.58 \times 10^{-3}$. B) Erodibility according to the extended model (eq. 11). The fit gives an adjusted $R^2$ of 0.375 and a prefactor of $9.66 \times 10^{-3}$.**





### 4.2.3 Further controls on erodibility

While the extension of the model provides some improvement when compared to our data, the fit is far from being satisfactory. We conclude that other rock properties that are not investigated here – for example, microstructure, composition and mineralogy, the presence or absence of a matrix, grain size, or the grain boundary shape – exert a strong or even dominant control on the erodibility of rocks in fluvial impact erosion. Assuming that fractures preferentially occur along grain boundaries, we suggest that mineral grain size plays an important role, as has been previously put forward by Hobley (2005). Specifically, we can assume that only in a narrow area around the impact location, the deformation of the rock is strong enough to yield tensile stresses sufficiently large to cause fracture. The extent of the deformed zone with sufficiently high stresses can be assumed to scale with the size of the impactor $D$, and the fracture behaviour is controlled by the fraction of area within this deformation zone that is occupied by grain boundaries. We assume that for a given type of rock, the width of the weak zones along the grain boundaries is small in comparison to the diameter of the mineral grains. The relevant dimensional group for the problem is the ratio of mineral grain size $d$ and impactor size $D$, $d/D$. We expect that the erosion rate described by eq. (11) is further modified by a dimensionless function $G(d/D)$, such that

$$E = \frac{G(d/D)}{k_\zeta} \frac{v^2 Y_i Y}{(Y_i + Y)\sigma_T{}^2} I_{BL} E_{kin}$$

(12)

We can identify two competing effects of the relative size of impactor and mineral grains. First, the fraction of energy delivered to a particular area element of the boundary decreases with the number, total length or total area of grain boundaries on a unit surface area. The more boundaries are present within the deformation zone, the less energy a particular boundary will receive when an impact occurs. In this case, we expect that $G$ scales with the area of the boundaries, i.e., $G \sim d/D$. Second, as mineral grains increase in size in comparison to the zone of deformation, the probability that a grain boundary is directly hit by an impact decreases. In the limiting case, if the impact hits in the centre of a very large grain, the deformation at the grain's boundaries may be too small to cause damage. In this case, we expect that $G$ scales with the likelihood of the impactor hitting on or close to the boundary, i.e., inversely with $d/D$, implying $G \sim D/d$. Consequently, we expect that $G$ is a humped function with a maximum at an intermediate value of $d/D$. We will further develop this concept in a separate paper.

### 4.3 Application of the laboratory experiments to natural rivers

The erosion rates measured in the mills are proportional to erodibility, since erosivity was held constant. However, absolute values for erosivity are not known, so we obtained only relative information on erodibility. Here, we suggest two theoretical frameworks to scale up relative erodibility values from the process scale to the spatial and temporal scales of channel evolution. These are based on (i) erodibility, energy delivery and stream power (Section 4.3.1), and (ii) explicit upscaling of sediment-flux-dependent erosion laws to long time scales (Section 4.3.2). We then touch upon the implication of erodibility on the



channel long profile in both models (Section 4.3.3). Finally, we discuss the application of the measurements to plucking, the other common erosion process in natural rivers (Section 4.3.4).

### 4.3.1 Erodibility, energy delivery, and stream power

The stream power incision model (SPIM) states that fluvial erosion rates are an increasing function of stream power (e.g., Lague, 2014; Seidl et al., 1994). It is routinely used to model the long-term evolution of river systems in mountain regions (e.g., Barnhart et al., 2020). Most commonly, the SPIM is written as

$$E = k_e A^{\hat{m}} S^{\hat{n}}.$$

(13)

Here, $E$ is the erosion rate, $S$ is the channel bed slope, $A$ is the drainage area, and $\hat{m}$ and $\hat{n}$ are dimensionless constants. The scaling factor $k_e$ is often referred to as the erodibility, but also subsumes effects other than rock property controls, such as hydrology, channel morphology, and sediment supply (e.g., Gasparini and Brandon, 2011; Lague, 2014).

We have already used the linear dependence in impact erosion of erosion rate on the energy delivered to the substrate to obtain theoretical relationships between erodibility and rock mechanical properties (see Section 4.2, eq. 1; Bitter, 1963; Engle, 1978; Sklar and Dietrich, 2004). Such a linear relationship has also been suggested for other erosion and fracture processes of brittle materials (e.g., Brantut et al., 2014; Cerfontaine and Collin, 2018). As before (eq. 1), the erosion rate can in this case be written as the product of erosivity $\chi$, the amount of energy per unit time and area that is transferred to the rock, and erodibility $\zeta$, which describes the rock's response to energy input. This idea provides a direct connection to the SPIM. Stream power per unit width $\omega$ describes the maximum amount of energy available in the river per unit area and time, and thus has the same units as erosivity $\chi$. Generally, only a small fraction of this energy is used for bedrock erosion (e.g., Turowski et al., 2013). In a revised version of the SPIM, erosivity can thus be expressed as the product of unit stream power $\omega$, and a dimensionless factor $a$ that quantifies the fraction of energy used for erosion. The latter can take values between zero and one. The erosion rate is then given by

$$E = \zeta a \omega.$$

(14)

The fraction of energy available for erosion $a$ can be expected to depend on site-specific parameters, including channel morphology, discharge, its variability, sediment load, or stream power. Assuming that variables $a$ and $\omega$ are independent of erodibility $\zeta$, equation (14) implies that the erosion rate in natural systems is proportional to erodibility $\zeta$ within the stream power paradigm. The relative values measured in the mills can thus be applied to natural stream systems.

### 4.3.2 Explicit upscaling of sediment-flux-dependent erosion laws

Turowski (2021) explicitly upscaled a sediment-flux-dependent erosion law of the form

$$E = K \zeta \frac{Q_s}{W} (1 - C)$$





(15)

to long time scales by integrating over the distribution of water discharge. Here, $K$ is a constant of proportionality, $Q_s$ is the sediment supply, $W$ is the channel width and $C$ the fraction of the channel bed covered by sediment. The upscaled, long-term erosion rate $\overline{E}$ is given by

$$\overline{E} = K\zeta F \frac{\overline{Q_s}}{W}.$$

(16)

Here, $\overline{Q_s}$ is the long-term mean sediment supply, and $F$ is a dimensionless function that depends on climate, channel geometry and bedload transport dynamics (e.g., the threshold of motion). Again, from eq. (16), it is clear that the long-term erosion rate in natural systems is proportional to erodibility $\zeta$. The relative values measured in the mills can thus be applied to natural

stream systems.

### 4.3.3    Implications for the channel long profile

Despite the agreement in the dependence on erodibility of long-term erosion rates in the revised SPIM (eq. 14) and a sediment-flux-dependent incision model (eq. 16), both erosion laws lead to contradicting predictions for erodibility-dependence of the channel long profile. Both the revised SPIM and the upscaled sediment-flux-dependent incision model of Turowski (2021)

predict a steady state channel long profile of the form

$$S = k_s A^{-\theta}.$$

(17)

Here, $\theta$ is the concavity index, and the steepness index $k_s$ depends on discharge variability, channel geometry, and sediment dynamics in the channel in the upscaled model of Turowski (2021), and on erosion rate and $k_e$ in the standard formulation of

the SPIM (see eq. 13). In the revised SPIM (eq. 14), eq. (17) becomes (Appendix A)

$$S = k_s{}' \left( \frac{E}{a\zeta} \right) A^{-\theta'}.$$

(18)

Here, the steepness index $k_s{}'$ and $\theta'$ are constants (Appendix A). From eq. (18), in the revised SPIM, channel bed slope in a steady state channel is inversely proportional to erodibility $\zeta$, e.g., if erodibility $\zeta$ is reduced by a factor of ten, slope is expected

to increase by a factor of ten.

For the sediment-flux-dependent incision model, Turowski (2021) derived an explicit solution for eq. (17). Here, we use a simplified version of this solution, assuming that bedload transport rates are independent of channel width, implying $q = 0$ in Turowski's (2021) notation. This seems to be a common observation in natural systems (e.g., Schoklitsch, 1934; Rickenmann,

2001). In this simplified version, the slope-area relationship becomes

$$S = k_s{}'' \overline{E}^{\frac{1}{n}} A^{-\theta}.$$



(19)

Here, $n$ is the slope exponent in the bedload transport equation, which typically has a value of 1.5 to 2 (e.g., Rickenmann, 2001, see also the discussion of Turowski, 2018). The steepness index $k_s''$ is a function mainly of discharge variability, channel cross section geometry and the threshold of motion (see Turowski, 2021, for more details). In this simplified model, channel slope is independent of erodibility. Even though the model of Turowski (2021) permits some other solutions that introduce an erodibility dependence into the slope-area relationship, this dependence is, in general, much weaker than linear.

### 4.3.4 Potential application to plucking

In our erosion mills we simulated the process impact erosion, which is often termed abrasion in the fluvial bedrock erosion literature. Next to impact erosion, fluvial plucking is a common erosion fluvial process (Whipple, 2000b). Plucking consists of the mobilization of bedrock particles larger than pebble size (medium diameter >4mm), which are detached from the bedrock by fracture propagation in situ. It can be a dominant erosion process in some river environments instead of abrasion, especially in highly jointed and fractured rocks (e.g., Beer et al., 2017; Bretz, 1924; Dubinski and Wohl, 2013; Whipple et al., 2000b). Here, we briefly discuss conceptually when the laboratory-derived values can be used to describe erodibility in the plucking process. Chatanantavet and Parker (2009) conceptualized plucking as a two-stage process including (i) the production of pluckable blocks and (ii) their mobilization by the flow. In the block production stage, cracks need to expand until a block is completely detached from the bedrock. This can happen by chemical and physical weathering, either of which can be the dominant process of block production in natural settings. When crack propagation by physical weathering is driven by the impacts of moving bedload particles, it is termed macro-abrasion. Using simplified laboratory experiments, Beer and Lamb (2021) demonstrated that the amount of fine and coarse erosion products fall on the same general trend with impact energy normalized by the square of the tensile strength of the rock. This indicates that the geotechnical controls on erodibility are the same for both processes of impact erosion and macro-abrasion. Beer and Lamb (2021) also identified an energy threshold as the transitory regime between impact erosion and macro-abrasion. Whether and how these laboratory-scale investigations translate to natural environments is currently unclear. However, from the available results, we expect that the relative erodibility measured in our mills is representative also for systems where erosion by plucking dominates, and in which macro-abrasion processes – in contrast to chemical weathering or pre-existing tectonically formed joints and fractures – lead to the formation of pluckable blocks.

## 5 Conclusion

We have extended the theoretical description of erodibility in the process of fluvial impact erosion, and tested it against data raised in dedicated experiments to measure relative erodibility and geotechnical properties of the rock. Geotechnical parameters such as compressive and tensile strength, Young's modulus, density and Poisson's ratio strongly co-vary, preventing a purely empirical evaluation of the geotechnical controls on erodibility in fluvial impact erosion. We therefore





assessed our data in the context of the brittle fracture theory suggested by Sklar and Dietrich (2004), and extended this theory with physically-based arguments. In addition to Young's modulus and fracture strength of the substrate, as had been previously
suggested (Sklar and Dietrich, 2004; Beer and Lamb, 2021), we reason that erodibility depends on the substrate's Poisson ratio, its mineral grain size, and Young's modulus of the impactor. We provide two theoretical framework where the relative erodibilities measured in our mills scale linearly to field situations, based on (i) a revised stream power incision model, and (ii) on a sediment-flux-dependent incision model including the tools and cover effects. As such, the relative erodibilities measured in the laboratory can be applied to scale erosion rates over long time scales. However, both approaches lead to
contrasting predictions regarding the dependence of channel bed slope, and thus channel long-profile, on lithology. In the revised stream power model, slope is inversely proportional to erodibility. Given that erodibility varied over nearly six orders of magnitude even for the limited range of rock types investigated in this study, this prediction implies a strong dependence of channel bed slope on rock properties. The sediment-flux-dependent model predicts an independence of or at most a weak dependence of channel bed slope on erodibility, which arises due to the self-organisation of the river channel in an erosional
steady state. These contrasting predictions may provide a convenient way of testing various models against each other using field data.





## Appendix A: Slope-area relationship in the revised SPIM

In the revised SPIM (eq. 14), erosion rate is given by

$$E = \zeta a \omega.$$

(A1)

Unit stream power is defined by

$$\omega = \frac{\rho g S Q}{W}.$$

(A2)

Here, $\rho$ is the water density, $g$ the acceleration due to gravity, $W$ the channel width, $Q$ a representative water discharge, and $S$ the channel bed slope. Width is assumed to scale with water discharge (e.g., Leopold and Maddock, 1953)

$$W = k_W Q^b.$$

(A3)

Here, $k_w$ is a dimensional coefficient and $b$ a dimensionless constant with a value of $b \approx 0.5$. Likewise, discharge is related to drainage area $A$ by (e.g., Seidl et al., 1994)

$$Q = k_Q A^c.$$

(A4)

Here, $k_Q$ is a dimensional coefficient and $c$ a dimensionless constant. Substituting equations A2, A3, and A4 into A1 yields


$$E = \rho g \zeta a \frac{k_Q^{1-b}}{k_W} S A^{c(1-b)}.$$

(A5)

Solving eq. A5 for slope gives

$$S = k_s' \left(\frac{E}{a\zeta}\right) A^{-\theta'},$$

(A6)

with

$$k_s' = \frac{k_W}{\rho g k_Q^{1-b}},$$

(A7)

and

$$\theta' = c(b - 1).$$

(A8)

Note that $a$ possibly depends on discharge, slope or sediment supply. Taking this dependence into account would change the scaling exponent in the relationship between slope and drainage area.



**Competing interests**

The authors declare that they have no conflict of interest.

**Acknowledgments**

We thank Markus Reich for constructing the erosion mills and overseeing their production. Benjamin Huxol helped with sample preparation and experiments. The NAGRA drilling team provided drilling expertise to obtain the rock cores. Quarry operators and local authorities are thanked for the permission for field sampling. Michael Naumann operated the machines for

the geotechnical measurements, and Brian Brademann, Stefan Gehrmann, Nicolai Klitscher, and Matthias Kreplin supported sample preparation in the workshop. We thank Ismail Albayrak and Michelle Müller-Hagmann for sharing their data compilation of experimental abrasion rates, and Claire Masteller and Ron Nativ for discussions. This study was funded by NAGRA.

**Data availability statement**

All data are available through the publication by Pruß et al. (2023).

**Author contributions**

JMT designed the study, performed the data analysis, developed theory and interpretation, and wrote the paper. GP developed

experimental protocols, coordinated sample collection and preparation, conducted the experiments, and post-processed and curated the data. AV and AB advised on and helped with geotechnical measurements. ALa, ALu and FK advised on and assisted with sample collection, geological characterisation and stratigraphy. All authors contributed through discussion and to the writing of the paper.

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
