# Peer review of "Geotechnical controls on erodibility in fluvial impact erosion"

_EGUsphere, 2023_

## Author Comment (AC1)

We thank the reviewers for their constructive comments. Our replies are given in *italics*.

Reviewer #1
The submitted manuscript presents a study designed to explore the controls on rock erodibility. The manuscript is interesting and well written. I recommend publication after the following minor comments have been considered.
*Thanks for the constructive comments.*

Line 12: "uniaxial compressive strength"
*Changed*

Line 13: "Poisson's ratio"
*Changed*

Line 21: "presents"
*Changed*

Line 93: So, these discs were 191 to 193 mm in diameter? Perhaps this should be stated here?
*That the cores have a diameter between 191 and 193 mm is already stated in the previous paragraph (line 77).*

Line 95: What lithology is the Passwang Formation? Perhaps this should be stated here?
*Sandy limestone; added now, as well as the unit numbers. We had the problem of large quartz veins in our core, which caused to fracture them into pieces with a length smaller than 120 mm. Unfortunately, we did not notice this in the field. See also Table 1.*

Line 104: The samples were dried following preparation? If so, how were they dried?
*After cutting, samples were sitting around in the lab for a couple of months before testing. A specific drying procedure was not applied. But it could be said that they were air-dried at room temperature. We added a sentence in section 2.3 on geotechnical measurements.*

Line 136: What does LDPE stand for?
*Low density polyethylene, added now.*

Line 137: How was trapped air removed exactly? Typically, fully saturating samples in the laboratory requires a vacuum pump. How are the authors sure that their samples were completely saturated?
*Trapped air was removed by pushing out bubbles, the sample bags were not vacuum pumped. We regularly weighed the wetted samples and assumed saturation when the weight did not change in subsequent weighings. We were not entirely sure that the rocks were completely saturated (and are not aware of a direct method to test this). Especially for the crystalline rocks, water uptake was slow and they were watered for several months before commencing experiments. Some more information on the wetting procedure is available in our method paper (Turowski et al., JHE 2023, in press, as cited in the manuscript).*

Line 141: What type of glass were the beads made from? What was the diameter of the glass beads? They are spherical?
*Spherical glass beads with a diameter of 6 mm. Added now. Specifics of the glass and material properties can be found in Turowski et al., 2023 in press.*

Line 149: Did any material spall from the samples during rinsing? Was this material collected?
*We did not observe spalling during rinsing. All solid material from the mills was collected and taken into account for the erosion rates.*

Line 161: This sentence suggests that bulk sample density was measured using an MTS load frame. Reword?
*Good point; reworded.*

Line 164: I think it's important to clearly state, if true, that these measurements were performed on dry samples. The strength of rock is typically lower when saturated with water. Since this "water-weakening" varies from rock type to rock type, it could explain some of the scatter in the experimental data (since the erosion tests are performed on wet rock and the petrophysical parameters are measured on dry rock).
*Good point, now stated in the introductory paragraph.*

Line 165: I assume this is the dry bulk density? How did the authors dry their samples prior to the measurement of dry mass?
*Samples were sitting in the lab for several months after cutting and before testing. Now stated in the introductory paragraph.*

Line 172: By "conversion rate", do the authors mean "displacement rate"? Can the authors also quote the strain rate here too? How was the displacement measured?
*We mean the rate at which the piston advances (in L/T). We use 'displacement rate' in the manuscript now, and define it as the rate of piston advancement.*

Line 174: Surely this is the stress, not the pressure? Unless the authors are talking about the pressure inside the piston that is used to calculate the force, which is in turn used to calculate the stress using the sample radius?
*Fair enough, changed.*

Line 176: Why would you use a constant force? Do the authors mean a constant force rate?
*Yes, constant force rate. Changed.*

Line 177: By "convergence speed", do the authors mean "displacement rate"?
*Changed, see comment to line 172.*

Line 178: "before fracture" is ambiguous. The authors mean the maximum force obtained immediately prior to the formation of the tensile fracture?

*Essentially, we used the first peak. In the tensile strength measurements, frequently several stress peaks were observed, with a drop-off n stress after the formation of the tensile fracture, and then another build-up of stress. Changed to 'maximum stress recorded before the stress drop due to failure.*

Line 181: The authors are sure that the behaviour is elastic at 50% of the UCS in all the experiments?
*The tangent moduli were taken at 50% of the UCS value in order to be outside of the recompaction domain and before the plastic deformation domain. It was decided to use the same value for all rock types in order to be able to compare all elastic moduli from all rock types. Given the obtained distribution of values, this assumption seems valid for our purpose.*

Line 182: "Poisson's ratio"
*Changed.*

Line 188: The authors are sure that the behaviour is elastic at 50% of the UCS in all the experiments?
*Same answer as to the comment on line 181.*

Line 212: What about bedding orientation? Did this influence the erosion rate?
*We mostly cored perpendicular to bedding directions. We have a few samples cored parallel, but we did not test them for the present study.*

Line 244: Poisson's ratio and UCS do not look particularly correlated (Figure 5d).
*We added a table with all correlation coefficients and slightly changed the wording.*

Figure 5: "Poisson's ratio"
*Changed, also in Fig. 6.*

Figure 5: Why were these cross-plots chosen? I'm not suggesting that the authors should plot every combination, but why do the authors only show how tensile strength, Young's modulus, Poisson's ratio, and density vary as a function of UCS?
*We decided to UCS as a benchmark for all correlations. It is obvious from the plot that, for example, tensile strength and Young's modulus are also correlated. We now added a table with all rank correlation coefficients.*

Line 245: Can the authors explain Kendall's rank coefficient? How is this calculated?
*This is a standard statistical measure. It is similar to Spearman's rank correlation coefficient in the sense that it does not require an a priori assumption on the type of relationship (contrary to Pearson's correlation coefficient, which assesses a potential linear relationship). The statistic is calculated by comparing all data pairs with each other and assigning a value of 1 if there is a positive step in the y-value, -1 if there is a negative step, and 0 if they are equal. Finally, all estimates are normalized to the range between -1 and 1. Kendall's tau is thought to be more robust than Spearman's coefficient, especially for small datasets.*

Line 259: How can the authors explain the large differences in erosion rates between samples cut from the same core (e.g., the data on Figure 4a)? This cannot be explained by differences in grain size etc.
*We cannot fully explain the differences at the moment. See the discussion.*

Figure 7: These data were collected using the same (similar) impactor material and geometry (shape and diameter)? If not, is it not useful to state these parameters in the figure caption?

*They mostly used natural pebbles, apart from Helbig et al., who used steel balls. Added to the figure caption. Note that the flow conditions, experimental set ups and total sediment mass also varied between these experiments. Our experiments are close to those of Sklar and Dietrich, 2001, with the same conditions except for particle shape.*

Line 265: Tensile strength can also be influenced by pore size and shape, as shown in Heap et al. (2021). Although this study focusses on volcanic rocks, the modelling results are relevant for other rock types, such as porous sedimentary rocks.

*Good point, added to the list.*

Heap, M. J., Wadsworth, F. B., Heng, Z., Xu, T., Griffiths, L., Velasco, A. A., ... & Deegan, F. M. (2021). The tensile strength of volcanic rocks: Experiments and models. Journal of Volcanology and Geothermal Research, 418, 107348.

*We did not add this publication; since we measured tensile strength directly, the empirical values implicitly account for the mentioned effect.*

Line 269: If the impactor type/size is important, I think this information should be provided in the figure caption for each of the datasets.

*This has been added to the figure caption.*

Line 276: Perhaps the authors should/could cite a few papers here that have previously demonstrated these relationships for rocks?

*We added two references, Horsrud, 2001, and Chang et al., 2006. The latter provide a compilation of earlier data. The passage now reads:*

*"For the rocks tested here, compressive strength, tensile strength, density, and Young's modulus are all correlated to each other (Fig. 5, Table 2) and to erosion rate (Fig. 6), as has been previously reported for other rocks (e.g., Chang et al., 2006; Horsrud, 2001). Correlation strength as measured by Kendall's τ rank correlation coefficient are similar to each other (Table 2)."*

Line 302: The "elastic modulus" is the Young's modulus?

*Yes, changed.*

Line 308: This constant does not therefore depend on rock type or rock properties? This has been previously demonstrated?

*In the contrary, it has been demonstrated and argued both from observations and theory that it incorporates rock properties that are not yet accounted for in models. This observation was one of the motivations for our study. We develop a more detailed description later in the paper.*

Figure 9: Do these plots show the same experimental erosion rate data?

*Yes, the same data as in the previous figures, framed within the formulation of the theory.*

Line 355 and elsewhere: The authors often refer to a "linear fit", which is actually a power law. Is this not misleading?

*It's a power law with an exponent of one, so linear.*

Line 363: Also pore size and shape.

*Added.*

Line 383: Unless I'm mistaken, I don't think the size of the impactors (the glass beads) used in this study is stated in the manuscript. Surely, based on this paragraph, it's very important for future studies to clearly state the impactor size?

*This is an excellent point. The impactor size of 6 mm is now stated in the method section.*

Reviewer #2

This is very impressive work on the experiments of the controls of rock properties on erosion rate. I suggest major revision.

*Thanks for the constructive comments.*

I have two major comments. 1) there needs to be much more details on the results. Currently the results section is so short, especially, when it comes to the interesting results of figure 5 and 6, there is less than 100 words! The authors choose different rock types, and different rock from the same sample for this impressive experiment. They have so much information that is missing here.

*We have added a table giving all rank correlation coefficients, and a few details on the observations. Since we develop most of our interpretation within the bounds of current theory in the discussion section, we do not think there is a lot more to say at this point.*

2) The discussion section can go much deeper, and there is a disconnection between results and discussion. The authors test different rock types, and found the erosion rate vary a lot. The discussion should focus on explaining the variation of results and the implications. With the new two theories, the authors can apply two theories to the results, and try to explain the scatter more specifically and carefully, rather than just saying there is a better trend or the theory probably works better with no proof.

*We actually think that we do what the reviewer suggests here: we discuss the variation in the data using available models from the literature and physics-based reasoning. We are unsure which two new theories the reviewer is referring to here. We compare the only model from the literature (Sklar and Dietrich, 2004), with an extension of this model presented in our paper. We do show that the new model gives an improved description of the data. The goodness of fit, measured by the R2, increases by 53% when using the new formulation. This is not nothing, yet, we agree that we do not fully understand the physics. This is fully acknowledged in the paper. Even though there is still a lot to learn, we think we made significant progress both in the quality and extent of the data and in the theoretical description. Further, we provide direct suggestions for further investigations, as well as testable predictions on the landscape scale.*

Also without a solid theory or solid explanation, expanding the results to bedrock rivers is a big stretch.

*We disagree with this point of view. We think the sections on upscaling and application to natural systems are essential parts of the paper. Contrary to the statement of the reviewer, we present arguments for the connection, rooted in currently available theory. We do not see how this is not solid. The popular stream power model predicts that variation in slope in natural channels should vary directly with erodibility, and this is not the case (we do not see large variations in slope at lithological boundaries). Note the arguments we present on upscaling are independent of the precise formulation of the erodibility, and the controls on it. The empirically measured values from the mill experiments can be used to obtain relative erodibilities, without the need to refer to the rock geotechnical properties. It is beyond the scope of the paper to present and discuss all of the evidence. Instead, we derive testable predictions by combining our experimental observations with theoretical predictions.*

Individual Comments:

  Line 35-36: "In the field,..., poor exposure", this is a confusing sentence, something is missing here.
*Replaced "they" with "such datasets" for clarification.*

  I found it was difficult to interpret the results shown in Table 1, Figure 3 and Figure 4. The axis of Figure 3 tells me nothing about the rock, I have to go back to Table 1 for every sample id and unit id. The authors decided to used locations as IDs of individual rocks, but as a reader, I am more interested in results from different rock types, rather than different locations. I suggest to use rock type as a primary id, and location and others as a secondary one, to differentiate them.
*We have added a colour code to Figure 3.*

[Figure]

**Figure 3:** Measured mill erosion rates for all samples (A) and lithological units (B). Gray boxes show the median (central horizontal line), and the 25th and 75th percentiles (bottom and top of the box). White squares show the mean, and whiskers the maximum and minimum erosion rates. The display follows the order of Table 1 (core ids, numerical unit id). Colouring denote lithology classes. Sample ids are composed of the leading core id, followed by a letter indicating the position of the sample within the core. Measured erosion rates vary over approximately six orders of magnitude across the different lithologies.

Line 141-145, how much is the glass bead abrasion and what is the difference of glass bead abrasion for weaker and harder rock? For a harder rock, more glass bead will probably be eroded, while not much for a weaker rock. Also more time is ran, more glass beads can be eroded. Given that the experiments use different rock types and run time, it is expected there will be different amount of glass beads that got eroded. If the glass bead gets eroded a lot, it will directly influence the erosion rates on the rock, which means the erosion rate is not solely controlled by rock properties here. The authors need to add more information here, and also potentially in the discussion since impactor properties influence $k\zeta a$ (Sklar and Dietrich, 2004).

*We added more information on the glass bead abrasion, including some statistics, both in the method section and the results. In the method section, the passage now reads:*
*"As abrasive tools in the mills, we used spherical glass beads with a diameter of 6 mm, originally designed for the grinding of pigments. For each experiment two independent bead sets of 150 g each were prepared to run in alternation. To keep track of bead abrasion, after each run the bead set was oven-dried for 24 hours at 40°C, and weighed to a precision of 0.01 g to obtain the total weight of the bead set. Wear was compensated for by exchanging glass beads or adding new ones. If one or several beads abraded to a diameter less than 5.6 mm, i.e. the mesh size of the sieve used to separate the beads after each run, the complete bead set was replaced by new beads."*
*In the results section, the passage now reads:*
*"Mass loss of the beads was mostly negligible, varying between 4×10-3 g and 8.9 g in 182 runs, with a mean of 1.2 g, a standard deviation of 1.7 g, and a median of 0.4 g. Mass loss of the beads exceeding 1.5 g (1% of the total bead mass) was observed for 43 runs. High mass loss of the beads was associated with slowly eroding rocks, due to a combination of higher bead abrasion due to rock strength contrasts and long run times. For most of the lithologies, mill erosion rates were comparable over the six runs (Figs. 3A, 4). Slowly eroding rocks showed higher variability (Fig. 3, see also Turowski et al., 2023), probably due to smaller total eroded volumes, and larger relative change in bead mass due to longer run times in comparison to quickly eroding rocks."*

Line 140-150. I am confused with the experimental runs. "For each experiment two independent bead sets of 150 g each were prepared to run in alternation.", so this means each experiment has more than 1 run? Then "Run duration was set depending on the erosion rates to between 4 hrs and 52 days to achieve a total mass loss of 1 - 10 g", this means only one run? And "To keep track of bead abrasion, after each run the bead set was oven-dried for 24 hours at 40°C", after how much time? Between 4 hrs and 52 days?
*Each experiment consisted of six runs. This was stated at the start of the following paragraph (old line 145). We have now moved the sentence to the start of this paragraph.*

Line 209. I cannot see how figure 5 give uncertainties, figure 5 shows rock properties, not erosion rates.
*Typo, should be figures 3 and 4. Corrected now.*

Line 241-245, "All measured rock geotechnical properties are correlated," it seems all properties correlate with compressive strength based on figure 5, rather than they are all correlated.
*It is unnecessary to display all combinations here. The obvious and strong correlations of, for example, both tensile strength and Young's modulus with compressive strength imply that they are correlated to each other as well.*

Figure 5 and 6 are the key results, but are only written in very short two sentences, less than 100 words! I highly suggest the authors to go into more details and guide the readers to the interesting results. For example, how erosion rate correlates with compressive strength, what is the best fit relation, and what is the comparison of each correlation.
*We have added a table with all correlation coefficients and a few more sentences. Our prime interest is in the overall correlations, rather than the structure of the data. These are interpreted in light of theory in*

*the discussion section. Within the frame of the paper, we think that all relevant points for the discussion are made here.*

Line 256-259, again, key results, but are difficult to get from the current use of ids.
*Colour code added to the box plots in Fig. 3B, see above.*

Figure 8, what value of rock resistance coefficient kζa is used here? And how did you choose the value? Is it possible that the remaining trends with compressive strength, tensile strength, Young's modulus, and density are due to kζa, especially when kζa captures controls on erodibility (e.g., Turowski et al., 2013; Auel et al., 2015) and also the glass beads (Sklar and Dietrich, 2004)?
*In both plots, the rock resistance coefficient was set to one (and could be calibrated from the fit). We have added this to the caption. As discussed in the text, the poor fit indicates that the coefficient is not constant, but varies with some unmeasured rock property.*

Line 340, why "It is related to f by the square of Poisson's ratio v"? There needs more explanation and references.
*This arises simply from indirect tensile stress and the definition of Poisson's ratio. Poisson's ratio is defined as the negative of the ratio of compressive and tensile strain, which is equal to the ratio of compressive and tensile stress in the linear, elastic part of the stress response. Since compressive strength appears squared in the equation, Poisson's ratio also needs to appear squared.*

Equation 3, there is kζa, but equation 11, there is $k_\zeta$. Are they the same, if not, what values?
*They are not the same, and their relationship is apparent from the comparison of equations 3 and 11. Essentially, kζa includes the dependence on Poisson's ratio, Y and Yi that is made explicit in the derivation in equations 4 to 11. So:*

$$\frac{k_\zeta}{k_{\zeta a}} \sim \frac{v^2 Y_i}{(Y_i + Y)}$$

Equation 7. I am confused here. In equation 4, you use f, but how does it change into fc in equation 7 when you use equation 4 to get equation 7. So if it is supposed to be fc in equation 7, all the equations after equation 7 and the results are supposed to change correspondingly.
*We use f to quantify the tensile elastic energy, which is relevant for erosion. Tensile deformation is indirect, due to the Poisson effect in reaction to the compressive deformation upon impact. Equations 5 and 6 are concerned with the compressive stress. Therefore, to relate them back to equation 4, we need to take into account the Poisson effect, which is done through the introduction of fc, defined in equation 8. The equations seem correct to us. We added a sentence with an explanation between equations 7 and 8: "The introduction of fc is necessary to take into account the Poisson effect connecting compressive deformation, which is treated in eqs. (6) and (7), to the indirect tensile deformation relevant for erosion (eq. 4)."*

Line 361-364, there is a strange conclusion. The experiment already measure a lot of properties, and the lack of fit is not because of the lack of measurements, but the lack of theory. Regardless how many things you measure, without a good theory, no one knows what to measure and to fit, not to mention how to fit.

*We agree in general with this statement. It is unclear to us how it relates to what we wrote and what the reviewer would like to see changed.*

Line 368-369, Why is assumed to scale with impactor size? What is the reference?
*Deformation seems to scale with impact energy. We added more details and a reference. The passage now reads:*
*"The extent of the deformed zone with sufficiently high stresses can be assumed to scale with impact energy (e.g., Wilson & Lavé, 2013), and thus with the size of the impactor D, and density. In the following argument, we focus on D, noting that the relationships should differ for different lithological groups (cf. Hobley, 2005). As such, the fracture behaviour is controlled by the fraction of area within this deformation zone that is occupied by grain boundaries."*

Line 369-370, what is the reasoning and reference of this assumption?
*This lies in the definition. If the boundaries occupy the majority of the volume, it would constitute the fabric of the rock, and the crystals or grains would be viewed as interbedded clasts without major relevance for the overall fracture properties. We now distinguish explicitly between matrix- and clast-supported rocks and have changed the text to:*
*"We assume that for a given type of rock (clast-supported rocks), the width of the weak zones along the grain boundaries is small in comparison to the diameter of the mineral grains. For matrix-supported rocks, the relevant grain size would be that of the matrix, rather than the clasts."*

It is a bummer that the paper stopped here, when readers go through the paper, find the current fit is not good, and think there is a better way, but do not have any idea of how better this will be. I suggest to connect this to the results of this paper in some way, and give a discussion of how this will explain the scatter in the current theory.
*We currently do neither have the data necessary to go into depth here, nor a convincing theoretical formulation. We have tried to obtain at least some preliminary data, but were not successful. The problem needs considerable more work. Yet, we think that the paragraph is important: it communicates our opinion on the dominant missing parameter, and provides clear expectations and testable hypotheses on it, using physics-based reasoning. We have added an illustrative figure and slightly rewritten the text.*

[Figure]

**Figure 10: Illustration of the effect of the rock's grain size $d$ on erosion rate. Fractures are assumed to preferentially occur along the grain boundaries. If the impactor size $D$ is much larger than $d$, the impact energy driving erosion is distributed to multiple boundaries, and the length of boundaries decreases with increasing $d/D$. Thus, the energy delivered per unit boundary length and therefore the erosion rate increases with increasing $d/D$. If the impactor size $D$ is much smaller than $d$, the likelihood of hitting a grain boundary and causing fractures – and thus the erosion rate – decreases with increasing $d/D$. As a result, the erosion can be expected to be maximised for an intermediate value of $d/D$.**

Due to the lack of good explanation and good fit of the results, it seems too much to apply them to natural bedrock rivers and other erosional processes in Section 4.3 and 4.4. Also the authors state that they can be applied to natural systems, but there are no indirect or direct measurements to support that. I suggest to cut sections 4.3 and 4.4, or shorten into one section. At the same time, I suggest to expand on section 4.2, focusing on using the current theory to explain the trend and scatter more specifically and carefully, and the new two theory to guide reads to understand and to explain the results, from the same rock type and the different rock types.

*We disagree; we think these sections are essential parts of the paper. Contrary to the statement of the reviewer, we present arguments for the connection, rooted in currently available theory. The popular stream power model predicts that variation in slope in natural channels should vary directly with erodibility, and this is not the case (we do not see large variations in slope at lithological boundaries). Note the arguments we present on upscaling are independent of the precise formulation of the erodibility, and the controls on it. The empirically measured values from the mill experiments can be used to obtain relative erodibilities, without the need to refer to the rock geotechnical properties. It is beyond the scope of the paper to present and discuss all of the evidence. Instead, we derive testable predictions by combining our experimental observations with theoretical predictions.*

---

## Author Response (AR2)

We thank the AE for his constructive comments. Our replies are given in *italics*.

AE comments
The Authors have made a good set of revisions to the reviewers suggestions. I have one point I feel should be addressed if possible under minor revisions. To me there seemed some ambiguity in the response to point 2 of reviewer 2 "R2: With the new two theories, the authors can apply two theories to the results, and try to explain the scatter more specifically and carefully, rather than just saying there is a better trend or the theory probably works better with no proof."

Authors Response: We actually think that we do what the reviewer suggests here: we discuss the variation in the data using available models from the literature and physics-based reasoning. We are unsure which two new theories the reviewer is referring to here.

*We interpret the comment of the reviewer to pertain to the treatment of the data, and therefore to a different aspect of the paper then the statement in the conclusion quoted by the AE. Let us explain. In sections 4.1 and 4.2, we discuss the data obtained in our experiments, in particular how geotechnical properties control the erodibility as measured in the mill experiments. There, we compare the data to theory. In contrast to the statement of the reviewer that we use "two new theories", we discuss the data in context of the state-of-the-art model from the literature (originally put forward by Sklar and Dietrich, 2004), and then extend this theory further using physics-based reasoning (section 4.2.2). In a final step, we argue that even the extended theory is insufficient to explain the variation in the data. In response, we suggest that the dominant missing control lies in the grain size and provide a qualitative argument on how this control should work and scale.*

*As stated in the rebuttal, it is unclear to us which "two new theories" the reviewer referred to here – we discuss an "old" theory, and extensions of it.*

*We do not think that reviewer refers to the upscaling arguments here (section 4.3). First, these are also not "new theories". Second, they cannot be applied to "the results […] to explain the scatter more specifically and carefully", as suggested by the reviewer. See below for further details.*

*We still think we do what the reviewer suggests – i.e., "apply two theories to the results, and try to explain the scatter". We (i) empirically compare mill erosion and geotechnical data (section 3.1), we discuss some general observations (section 4.1), concluding that a purely empirical approach does not yield any advances, and that the data need to be looked at in the context of theory (line 293-295). We do the latter by comparing the data to the existing state-of-the-art theory (section 4.2.1), and then extend it by using physics-based reasoning (section 4.2.2). Finally, we suggest that the theory is incomplete and suggest directions for further investigation (section 4.2.3).*

*As a final remark, we disagree strongly with the remark that we just claim that "there is a better trend or the theory probably works better with no proof". As explained above we evaluate the data within the context of theory to which we provide novel insights. We believe our analysis and discussion yields novel and significant advances.*

However in the conclusions: "We provide two theoretical frameworks where the relative erodibilities measured in our mills scale linearly to field situations, based on (i) a revised stream power incision model, and 525 (ii) on a sediment-flux-dependent incision model including the tools and cover effects. As such, the relative erodibilities measured in the laboratory can be applied to scale erosion rates over long time scales."

*This statement from the conclusion pertains to section 4.3 ("Application of the laboratory experiments to natural rivers"). There, we develop arguments for the scaling of erodibility in the theoretical frameworks of the stream power incision model and sediment-flux-dependent incision. We argue that in both frameworks, erodibility scales linearly for erosion rates. However, the two model frameworks lead to different predictions for the dependence of the channel long profile on lithology. The idea of the chapter is to provide a broader context of the applicability of our results. Please note a couple points on this:*

- *Neither of the arguments can be considered as a "new theory" – although we provide some new points of view (especially for the stream power model), we evaluate existing theories for this particular application. In our point of view, our points here have not been explicitly made in the literature before, but are implicit in published theory (e.g., Turowski, 2021, as cited in the paper).*
- *The upscaling arguments are completely independent both of the relationship between erodibility and geotechnical properties, and of the particular structure of the data.*
- *As such, it is not necessary to fully understand how geotechnical properties determine erodibility to apply them. The relative mill-measured relative erodibilities can be applied to erosion rates in natural rivers.*

*The latter is the main point that we want to make in the statement of the conclusion, quoted above, is that the measured erodibility values can be directly applied to natural rivers.*

I was not entirely convinced by the rebuttal offered by the authors here - there seemed to be a bit of contradiction in the response and the statement in the conclusion and I would invite the authors to have another look at their response to this issue - and perhaps either re-frame their response or provide some text in the paper that explains or makes this clearer? If (as stated in the response) you think you are actually doing what the reviewer suggested - perhaps that section could be re-examined and made clearer?

*We have implemented some revisions to clarify these issues:*
*We changed the final sentence of the introduction to (line 65)*
*"In addition, we discuss the broader application of the observed erodibilities, in particular their upscaling of the results to natural rivers within the two currently competing theoretical frameworks of the stream power model and sediment-flux-dependent bedrock erosion."*
*We added explanatory sentences to the opening paragraph of section 4.3:*
*"In this section, we put the observations on erodibility from our laboratory-scale experiments into a broader perspective." (line 418) and "We note that the arguments presented in this section are independent of the precise geotechnical controls on erodibility and the structure of the mill data." (line 425).*
*In the conclusion (line 521), we changed the statement quoted by the AE to*
*"We suggest that the relative erodibilities measured in our mills scale linearly to field situations, based on theoretical reasoning using both (i) a revised stream power incision model, and (ii) on a sediment-flux-dependent incision model including the tools and cover effects."*